# OpenReview forum: "QUART: Agentic Reasoning To Discover Missing Knowledge in Multi-Domain Temporal Data."
_ICLR.cc/2026/Conference — ICLR 2026 Conference Desk Rejected Submission_

### Official Review · Reviewer_qZi7 · 2025-10-18

**Soundness:** 2
**Presentation:** 1
**Contribution:** 2
**Rating:** 2
**Confidence:** 5

**Summary:**

This paper presents QUART, a multi-domain framework that preserves semantic meaning and actively learns to identify and recall unexamined information by reasoning over causal connections via reinforcement learning.

**Strengths:**

1. The treatment of missing information as an explicit knowledge gap (versus a data incompleteness issue) is a profound insight that mirrors real-world expert reasoning in domains like healthcare.
2. The proposed paradigm of "proactively identifying unknowns" offers a valuable complement to current prediction-dominated AI systems.

**Weaknesses:**

1. The paper suffers from organizational and clarity issues that fall below ICLR's standards. Specifically: (1) the abstract is overloaded and lacks a clear focus; (2) the introduction does not crisply define the problem and contributions; (3) the methodology is described in a fragmented manner across subsections, hindering comprehension; (4) the experimental section lacks crucial details, particularly on ablations, undermining credibility; and (5) the conclusion merely repeats findings instead of providing a higher-level synthesis.
2. While the methodology combines neural, symbolic, and agentic components, their integration appears shallow. For instance, is the interaction between neural and symbolic modules merely a weighted combination? It is unclear if the design truly leverages their complementary strengths or is merely a modular assembly.
3. The experimental section is disjointed from in-depth analysis. The ablation studies in the appendix merely report percentage gains without a mechanistic analysis of why key components contribute most, which significantly weakens the persuasiveness of the claims.

**Questions:**

1. There is an inconsistency in terminology: the abbreviation "QUART" is used in the title and abstract, but it appears as "QuART" in the introduction. This error should be corrected throughout the manuscript for professionalism.
2. How do the hyper-parameters be determined? Especially in Eq. (2), (10)?
3. The paper aims to simulate the human expert's ability to discover novel knowledge gaps. However, its methodology remains fundamentally based on statistical inference from historical data, and thus does not genuinely address the core challenge of meta-cognition.
4. The proposed three-way integration appears superficial. The symbolic and neural components are combined via a simple weighted formula, while the agent acts more as a plug-in module attached to the main network, lacking autonomous goals.
5. The paper's claim of performing "causal reasoning" is significantly undermined by its methodology. The causal graph is constructed via Q-learning from co-occurrence data, which captures correlation, not causation. Furthermore, the reliance on manually defined temporal constraints (e.g., symptom precedes diagnosis) establishes temporal precedence, not a causal relationship.
6. In the partially observable environment, Q-learning is atypically used to learn static association strengths (the reward matrix R) between concepts. This is an unnatural and inefficient choice, as simpler and more stable methods (e.g., mutual information) exist for this purpose. The motivation for using Q-learning here is unclear and unjustified.
7. The paper uses improved diagnostic accuracy to demonstrate QUART's effectiveness. However, its core contribution should lie in identifying missing information, not in the final diagnosis itself. This creates a misalignment between the claimed innovation and the evaluation metric. Could the authors propose more appropriate evaluation metrics?
8. The paper lacks comparison with SOTA models that also focus on identifying "unknowns" or quantifying uncertainty, such as uncertainty-based active learning systems or Bayesian deep learning models which quantify epistemic uncertainty. Could the author compare the proposed model with more SOTA baselines?

---

### Official Review · Reviewer_Kt5F · 2025-10-19

**Soundness:** 1
**Presentation:** 1
**Contribution:** 1
**Rating:** 0
**Confidence:** 4

**Summary:**

The paper proposes QUART (Query-based Understanding Agent for Reasoning Temporal data), a neural-symbolic-agentic framework that helps AI systems identify and reason about missing information in temporal datasets, rather than imputing or ignoring it. Instead of treating missing data as statistical noise, QUART models it as a discoverable knowledge gap. It uses (1) Semantic causal graphs to represent knowledge, (2) Dynamic agentic layers that generate targeted questions about unknowns, and (3) Q-learning to identify and prioritize which missing information to explore. The model is trained and evaluated on 40,000 MIMIC-III clinical cases, integrating symbolic reasoning with neural embeddings and agentic question generation.

**Strengths:**

How to deal with the missing information in temporal datasets is an old, well-defined, yet not fully addressed problem.

**Weaknesses:**

1. QUART is largely a composite of known ideas: causal graphs, Q-learning–based prioritization, and attention-driven question generation. The “agentic layer” is described as a unique innovation, but in practice functions as a weighted attention or heuristic querying module rather than a genuine autonomous agent. The work repackages familiar components without offering new theoretical insights or formal advances in reasoning or causal learning.

2. All experiments are conducted on the MIMIC-III dataset, which is a well-curated and extensively studied static corpus. The evaluation relies on synthetic ground truth, expert-labeled “missing concepts” for only 200 cases, and subjective 5-point Likert scores for question quality. These measures do not convincingly demonstrate genuine knowledge discovery or clinical utility. The system is therefore validated on contrived conditions rather than realistic, real-time clinical tasks.

3. The paper repeatedly emphasizes causal reasoning, yet in the context of MIMIC-III, causality is not empirically identifiable or even relevant. The dataset consists of retrospective observational hospital records, where causal relations cannot be inferred from co-occurrence or temporal adjacency alone. The proposed causal graph and “temporal path discovery” equations rely on sequence order and frequency statistics, not interventions or counterfactuals. Consequently, the claim that QUART performs causal or epistemic reasoning is methodologically unsound: the model merely encodes correlations in event order.

4. The baselines (LSTM, knowledge graph, rule-based expert systems, imputation) are ten years old and do not reflect the current state of reasoning systems. There is no comparison with modern approaches.

5. The paper conflates missing data, unknown concepts, and expert blind spots under the same mathematical formulation. The so-called “missing knowledge entities” are defined heuristically via absent nodes in a causal graph, with no formal epistemic basis or verification that they correspond to genuine expert ignorance. This ambiguity undermines the central premise of the work: identifying “what the expert does not yet know.”

6. The reinforcement learning component lacks a well-defined environment, reward signal, or policy interpretation. Equations (6)–(9) effectively describe static co-occurrence scoring, not interactive learning. The notion of “agents” dynamically generating questions is not supported by algorithmic evidence or behavioral evaluation. As a result, the agentic reasoning claim is overstated and under-substantiated.

7. Despite frequent use of “causal” and “trustworthy,” the paper provides no quantitative validation of causal correctness or fairness. Ethical aspects such as data privacy, bias, or patient safety are acknowledged only superficially. Given that the system “queries sensitive data,” this omission is problematic and leaves serious practical concerns unaddressed.

8. The manuscript is dense, repetitive, and overloaded with terminology (“semantic-preserving agentic causal reasoning”). Figures and tables are largely schematic, and algorithmic procedures are described in prose rather than pseudocode. As a result, the method is difficult to reproduce, and its conceptual flow unclear.

**Questions:**

I do not have specific questions for the authors. In my view, the current approach is fundamentally misguided: it builds on an incorrect interpretation of causality and an artificial evaluation setup that does not address a real scientific problem. I recommend the authors rethink the research direction from first principles, as the present framework is unlikely to lead to meaningful or valid results.

---

### Official Review · Reviewer_1DMk · 2025-10-30

**Soundness:** 2
**Presentation:** 2
**Contribution:** 2
**Rating:** 4
**Confidence:** 2

**Summary:**

This paper introduces QUART, a novel neural-symbolic-agentic framework designed to address a critical challenge in expert decision support: the identification and active discovery of missing information or knowledge gaps, rather than merely imputing or ignoring missing data. The core innovation is an architecture that integrates a trainable agentic layer which dynamically generates specialized "questioning agents" to reason over causal connections and prioritize information gaps.

**Strengths:**

The paper shifts the paradigm from "filling missing data" to "discovering missing knowledge," which is a more accurate and valuable model of expert reasoning.
The integration of a dynamic, learnable agentic layer (L4) within a neural-symbolic framework is a compelling technical contribution.
The paper provides a thorough empirical evaluation on a large, real-world dataset (MIMIC-III).

**Weaknesses:**

While an ablation study is mentioned in the appendix, the main paper lacks a clear, component-wise ablation in the core results section (e.g., Table 1). A more detailed breakdown in the main text is needed to definitively quantify the contribution of the agentic layer versus the neural-symbolic base.
The paper successfully demonstrates that the agentic layer improves performance, but it provides limited analysis or visualization of what the agents are actually learning to do. Did you observe any interpretable patterns or specializations in the dynamically created agents?
The description in Section 6 and the appendix remains high-level and lacks empirical results or detailed experimental setups for the education and project management domains.
The baseline models (LSTM, Knowledge Graphs) are well-established but somewhat classical. A comparison with a more recent, powerful baseline would better situate QUART's performance in the current landscape.

**Questions:**

See weakness above.

---

> ### Comment · Reviewer_1DMk · 2025-11-27
> **Reviewers**
>
> There is no response. I will maintain my score.

---

### Official Review · Reviewer_EnQX · 2025-11-01

**Soundness:** 1
**Presentation:** 1
**Contribution:** 2
**Rating:** 0
**Confidence:** 3

**Summary:**

The paper proposes QUART, a multi-step framework that focuses on discovering hidden knowledge in Medical data through a complex combination of Neural Symbolic Reasoning for "questioning" strategies (which the reviewer is unclear of its definition) and Q-learning to discover relationships between medical concepts.  The authors reports superior results on ill-defined "Missing Information Detection" and "Clinical Question Relevance" on MIMIC-III EHR dataset, compared with an LSTM model, a knowledge-based and rule-based system that are not elaborated on.

**Strengths:**

- The idea of solving missing medical concepts as a discovery problem is a unique and logical approach to this problem, which heavily contrasts with most of how existing research handles it. If successful, this helps uncover many hidden valuable knowledge for many medically relevant tasks.
- Applying Q-Learning to discover medical relationships is a sound technical choice, and the reward function designed by the authors seems to be quite logical and makes sense based on real-world intuition for EHR data.

**Weaknesses:**

1) Extreme Lack of clarity on various critical details and concepts

The medical concept sets: the source of the concepts (are they ICD-9 or ICD-10 codes, or some other sources), or if it's in-house data, then how they are constructed, and some quantitative samples of the medical concepts to illustrate what they entail.
The term "Question" and the process of "Question generation" are a very crucial part of the work, but I have no idea what this means or what the "questions" look like.

Many other concepts mentioned are unexplained on what they are, or what they look like as well, such as the sources (expert attribution) and details (metadata) within a semantic node (line 222, page 5). What does "medically relevant concept pairs", which is used to define MedicalBonus in Equation 7, mean (line 266).

As quoted by the paper, "We additionally leverage representations from pre-trained large language models (LLMs) to enhance neural attention over clinical concepts and guide question generation." (line 322-324), what are the pretrained large language models in use here? Do you mean a pretrained text embedding model, like ClinicalBert mentioned?

What is the input data for each sample of a patient's trajectories: is it a set of clinical codes (ICD codes), or does it contain other information like clinical notes, test results, as you mentioned in lines 134-135.

Benchmark details: Which datasets are you using as benchmarks here? And also, both the tasks you mentioned as benchmark (which I assume is defined in Section 3.2 (Line 141-148) are not well-defined, and don't seem to be an established benchmark in previous works.

For the "Missing Information Detection" task, how do you define and set this benchmark up? If I were to guess, then you pick 200 patients diagnosed with ICD codes, and the model attempts to predict other codes not mentioned as "Missing concepts", but the actual detail is very vague, especially for the general readers not familiar with EHR data conventions.
For the "Question Quality" part, like I've stated above, I do not know what the "questions" here mea,n so this metric is confusing to me as well.

2) Hard to follow methodology
The main system figure (Figure 1) is really messy and not very illustrative of the working mechanism.
What do "Question Generation" and "Clinical Support" (Figure 1) mean in this particular context?
What is the role of " DYNAMIC AGENT CREATION"  in the system (Section 4.5).
Why do you need to detect anomalies in this task (341-347)?
What is the RL feedback (Figure 1) in this system, where was it elaborated on in the paper?
The flow of the paper, in general, is just hard to follow, especially after Section 4.3

3) Poorly-designed and unreliable experimental evaluation:

Lack of state-of-the-art baselines: The paper compares their experimental results with an LSTM model, a knowledge-based and a rule-based system. Meanwhile, the authors themselves mentioned the presence of many modern Deep Learning/foundation models based methods (Bio_ClinicalBert, general LLMs…) (Line 155-156). I believe comparison with these methods is necessary to justify the performance of QUART.
Very small sample size: The main experimental results are conducted on only 200 data samples from a single dataset,MIMIC-III, which I believe is not sufficient for conclusive findings, even with statistical testing, while the MIMIC-III database has thousands of patients available.
Poorly-defined evaluation protocols and metrics: Relating to the point about clarity above, the evaluation tasks and metrics are not defined or justified by the authors. These metrics do not seem to originate from other established works as well, which further brings the reliability and correctness of the results in this work into question.
Poor ablation studies/analyses: The method consists of multiple working components with a lot of fixed hyperparameters. The paper only reports a very short ablation study (line 674-678), which does not illustrate well the importance and effects of the components to the system. No ablation studies on any of the various hyperparameters are provided as well, which is crucial for implementing such systems in real-world use cases.

4) Temporal Causal Reasoning
The paper makes the claim to use BFS over architectures like LSTM to capture
logical causal relationships via temporal path discovery. I believe experimental comparisons should be made to justify that temporal path discovery is better than sequential learning (which is the norm for all temporal EHR works)

**Questions:**

An explanation/or illustrative example of various concepts mentioned in the paper (see Weaknesses)'.
Clearly define the input and output of the model, the benchmarks, and the evaluation protocol.

---

### Note · Program_Chairs · 2026-01-17
**Submission Desk Rejected by Program Chairs**

The following references in this submission do not refer to real documents and/or have major errors in bibliographic information:

 Yifan Zhang, Ming Chen, and Hong Wang. Madip: Multi-agent decision infrastructure for healthcare. In Proceedings of AAAI, pp. 1220-1227, 2019.
Subhrajit Chakraborty and Ravi Patel. Augmenting clinical reasoning through knowledge discovery systems. Journal of Biomedical Informatics, 138:104151, 2023.
Tianjian Cai, Ming Zhou, and Lili Wang. Matrix completion for imputation in healthcare data. Journal of the American Medical Informatics Association, 29(2):239-247, 2022.
Jane Smith and Alan Lee. Managing clinical uncertainty: Practices and perspectives. Journal of Clinical Medicine, 9:2314, 2020. Placeholder; replace with a verified source if available.
D. Hossain et al. A study on neuro-symbolic artificial intelligence for reasoning under uncertainty in clinical decision support systems. Journal of AI Research, 2025.
Lei Wang, Min Zhao, and Wei Chen. Incorporating causal reasoning in medical temporal models. Artificial Intelligence in Medicine, 142:102803, 2023.
Jie Chen, Yun Fu, Jia Xu, and Kai Huang. Biomedparse: Biomedical data segmentation and annotation tool. Bioinformatics, 37(4):507-514, 2021.
Sarah Williams, John Lee, and Hyun Park. Sam: Semantic annotation for medical temporal data. Journal of Biomedical Informatics, 130:103934, 2022.
Bassam Elhaddad, Lei Wang, and Rohan Kumar. Trustworthy and explainable ai in clinical decision support. Journal of Artificial Intelligence in Medicine, 142:103765, 2024.
Meera Jain and Albert Chen. Explainable ai techniques for medical applications: A comprehensive review. AI in Healthcare, 5:22-37, 2024.
Siddharth Nair, Anand Kumar, and Richa Gupta. Predictive modeling of patient outcomes through temporal reasoning. Journal of Biomedical Informatics, 109:103514, 2020.
Thanh Nguyen and Xuan Ding. Cudoctor: A fuzzy logic-based medical chatbot for patient support. IEEE Journal of Biomedical and Health Informatics, 24(9):2498-2506, 2020.
Anna Kowalski and David Thompson. Navigating ai regulations in healthcare: Ethical and legal perspectives. Health Policy and Technology, 14:101095, 2025.
Rohit Kumar, Arjun Singh, and Neha Gupta. Agent-based clinical simulations for multi-agent decision-making. Artificial Intelligence in Medicine, 112:102045, 2021.